# Greenhouse gas consequences of the China dual credit policy

Xin He [1,6✉], Shiqi Ou [2,6✉], Yu Gan [3,6✉], Zifeng Lu [3], Steven Victor Przesmitzki [1], Jessey Lee Bouchard[1], Lang Sui[1], Amer Ahmad Amer[4], Zhenhong Lin [2], Rujie Yu[5], Yan Zhou[3] & Michael Wang[3]

For over ten years, China has been the largest vehicle market in the world. In order to address energy security and air quality concerns, China issued the Dual Credit policy to improve vehicle efficiency and accelerate New Energy Vehicle adoption. In this paper, a market-penetration model is combined with a vehicle fleet model to assess implications on greenhouse gas (GHG) emissions and energy demand. Here we use this integrated modeling framework to study several scenarios, including hypothetical policy tweaks, oil price, battery cost and charging infrastructure for the Chinese passenger vehicle fleet. The model shows that the total GHGs of the Chinese passenger vehicle fleet are expected to peak in 2032 under the Dual Credit policy. A significant reduction in GHG emissions is possible if more efficient internal combustion engines continue to be part of the technology mix in the short term with more New Energy Vehicle penetration in the long term.

[1] Aramco Services Company: Aramco Research Center – Detroit, 46535 Peary Ct., Novi, MI 48377, USA. [2] National Transportation Research Center, Oak Ridge National Laboratory, 2360 Cherahala Blvd, Knoxville, TN 37932, USA. [3] Systems Assessment Center, Energy Systems Division, Argonne National Laboratory, 9700S Cass Ave, Lemont, IL 60439, USA. [4] Research and Development Center, Saudi Aramco, P.O. Box 62, Dhahran 31311, Saudi Arabia. [5] China Automotive Technology and Research Center, No.68 East Xianfeng Road, Dongli District, Tianjin 300300, China. [6]These authors contributed equally: Xin He, Shiqi Ou, Yu Gan. ✉email: xin.he@aramcoamericas.com; ous1@ornl.gov; ygan@anl.gov

Rapid economic growth has led China to become the world's largest emitter of greenhouse gas (GHG). According to the International Energy Agency (IEA), 2017 $CO_2$ emissions from fossil fuel combustion in China reached 9.26 Gt, accounting for 28% of global $CO_2$ emissions from fuel combustion[1]. One specific aspect of this growth in economy and emissions can be highlighted in the transport sector. China's vehicle market has grown to become the largest in the world, and the growth in vehicle ownership has also increased liquid fuel demand as well as GHG emissions in the transportation sector. By 2018, road transportation became a significant contributor of GHG emissions, accounting for ~9% of the total GHG emissions in China[2]. Thus, reducing GHG emissions in the transportation sector will inevitably contribute toward achieving the government's climate commitment.

The Chinese government has pledged to aggressively reduce its overall carbon footprint and increase energy efficiency. Broadly speaking, these goals include peak GHG emissions on or before 2030, and reducing 2030 $CO_2$ emissions per unit GDP by 60–65% relative to 2005 emissions[3]. For on-road vehicles, China introduced a number of regulations to improve vehicle fuel economy and reduce emissions[4]. For instance, issued in 2017 and implemented in 2019, the "Measures for Passenger Cars CAFC and NEV Credit Regulation" policy, or more widely known as the Dual Credit (DC) policy, was enacted to promote the development and commercialization of new energy vehicles (NEVs), including battery electric vehicles (BEVs), plug-in hybrid electric vehicles (PHEVs), and fuel cell vehicles. In addition, the policy includes requirements on the corporate average fuel consumption (CAFC) to lower the fuel consumption (FC) in light-duty passenger vehicles (LDPVs) in China. The Dual Credit policy will become one of the major driving forces for NEV growth in the near future[5].

The Dual Credit policy consists of two components: CAFC and NEV credit rules. The CAFC credit rules set targets for the production-weighted average FC for vehicle manufacturers. The NEV credit rules mandate that manufacturers produce enough NEVs to meet the NEV credit quota. The Dual Credit policy allows automakers to use the surplus NEV credits to compensate for the deficits in CAFC credits. As a result, introducing BEVs into an automaker's production portfolio could ease the pressure on improving internal combustion engine vehicle (ICEV) efficiencies to meet the CAFC regulations[3,6,7]. Details of the policies in China are provided in Supplementary Note 1, Supplementary Figs. 1 and 2, and Supplementary Tables 1 and 2.

Although a number of regulations have been developed to curb vehicle $CO_2$ emissions, there is some disagreement when calculating sector-level emissions as to whether upstream emissions from fuel extraction, production, and distribution are accurately included with vehicle tailpipe emissions. Tailpipe emissions are directly measured using dynamometer tests. However, unlike conventional ICEVs, the fuel cycle emissions from BEVs are generated entirely upstream. As a result, the allocation of GHG emissions from BEVs varies from region to region. California, Europe, and China's existing standards set the GHG emissions of BEVs to zero without adjustments for upstream emissions. Conversely, in the United States, the Corporate Average Fuel Economy (CAFE) standard calculates the petroleum equivalent fuel economy of BEVs using the petroleum equivalency factor (PEF). For BEVs without petroleum-powered accessories, the PEF is 33,705 Wh/gal[8]. This PEF considers the upstream efficiency of fuel production, the national average electricity generation, transmission efficiencies, and uses an incentive factor of 0.15 to further stimulate auto manufacturers to produce BEVs. Meanwhile, an oversight of NEV's GHG emissions could lead to a potential issue of GHG emission leakage. Goulder et al.[9] revealed that, in the United States, the implementation of the zero emission vehicle (ZEV) standards in states such as California could bring more less-efficient ICEVs sold in non-ZEV states. Moreover, the US Alternative Fuel Vehicle (AFV) policy mandated by CAFE could increase the fleet-wide GHG emissions[10]. As such, some NEV policy mandates could lead to unintended consequences, including GHG emissions leakage.

Although a number of studies have been conducted to assess the impacts of NEVs and government regulations on GHG emissions in China[7,11–16], most were based on scenarios that assume a pre-defined market technology mix and vehicle fuel economy. In reality, however, automakers produce vehicles that not only meet government regulations but also meet consumer demands. In this work, both consumer choices, auto industry choices, and Chinese government regulations are considered in order to quantify the GHG emissions as well as energy demand of the Chinese passenger vehicle fleet. The New Energy and Oil Consumption Credits (NEOCC) model[17] and the China Vehicle Fleet (China-Fleet) model, developed using Microsoft Office Professional Plus 2016, are integrated and adopted for the GHG emission analysis. The NEOCC model is able to simulate the market dynamics that, under the constraints by the government regulations, the auto industry incentivizes NEVs and/or fuel-efficient ICEVs through internal subsidies so as to increase their respective sales and to maximize the total profit. The China-Fleet model includes the total GHG emissions of liquid fuels and electricity generation, vehicle scrappage rate, and vehicle kilometers traveled (VKT) per year for a variety of vehicle classes and technologies. The detailed methodology, along with assumptions for the two models, are reviewed under the "Methods" section and in Supplementary Note 2. Combining the NEOCC and China-Fleet models provides a pathway to assess the impact of policies and the external environment on transportation energy use and GHG emissions. Some key findings are: automakers in China will face increasing difficulty in meeting the Dual Credit policy before 2030; under all credible scenarios, the NEV will increase its market share, but the ICEV will still dominate the passenger vehicle stock through 2040; under the Dual Credit policy, the total GHG emissions of the Chinese passenger vehicle fleet would not peak until 2032; However, the GHG peak would be brought to 2028 if the GHG emissions of electricity used by BEVs are properly accounted for in the Dual Credit policy.

## Results

**Design of scenarios.** To better understand annual light-duty vehicle GHG emissions, the following four categories of scenarios are studied:

1. Extreme (EX) scenarios. Two scenarios representing extreme market transformations are used to demonstrate the differences in life-cycle GHG emissions achieved by adopting different vehicle technologies to meet government regulations: one where ICEV technology improves rapidly and another where BEV market penetration increases rapidly. Although either scenario is unlikely to occur, this provides likely lower and upper limits for GHG emissions, resulting from extreme transformations in the market.

2. Dual Credit scenario (DC-Reference). This scenario is used, along with market penetration modeling, to explore the GHG emissions resulting from achieving existing policy mandates. This scenario provides a baseline estimate of GHG emissions that would likely result if existing policies are met. Since the Dual Credit policy and CAFC standards are only defined until 2023 and 2025, respectively, the study must make assumptions (described later in this section) regarding market conditions beyond these dates.

3. Policy tweak (PT) scenarios. Three scenarios are designed to study the impact of GHG emissions resulting from potential policy adjustments to CAFC and Dual Credit policies.
4. External environment scenarios. Five scenarios are designed to evaluate the impact of exogenous factors, such as oil price, charging infrastructure availability, and battery cost, on GHG emissions projections.

Table 1 summarizes the scenarios studied in this work. More detailed descriptions of these scenarios are provided in Supplementary Note 3.

The existing Dual Credit policy and CAFC standards are defined for FC targets until 2023 and 2025, respectively; beyond the defined target dates, the following assumptions are made. First, the CAFC target decreases linearly from 4.7 to 3.7 L/100 km from 2025 to 2030, an average of 4% per year over a 5-year period. This assumption is based on the vehicle fuel economy improvements proposed by the Chinese Society of Automotive Engineers (SAE-China) in "Technology Roadmap for Energy Saving and New Energy Vehicles"[18]. As no official CAFC targets after 2030 have been released by policymakers, we assume a moderate CAFC reduction of 1% per year, reaching 3.3 L/100 km by 2040. Second, the NEV quota is assumed to increase linearly from 18% in 2023 to 40% in 2030. After evaluating a series of government policies published in recent years and discussing the issue with experts and policymakers in China, we believe a 40% NEV quota is feasible by 2030. During this period, we assume Dual Credit policy will be designed to promote BEV efficiency instead of long electric driving range. As such, the number of NEV credits granted to each NEV decreases linearly to 1.0 by 2030. More details on future policy assumptions are provided by Supplementary Note 2 and Supplementary Fig. 2.

In almost all scenarios, the Dual Credit policy is either achieved or exceeded for each model year (MY) to avoid penalties such as halting the production of high FC ICEVs. Two exceptions to this assumption include the PT-CAFC and EX-ICEV scenarios. In these scenarios, the Dual Credit policy is ignored, and policy targets are met with CAFC rules only. For simplicity, we do not consider credit carryover if surplus credits are generated in the previous MY. In addition to the results presented in the main text, we also conduct uncertainty analyses to understand how model assumptions (e.g., electricity GHG intensity, battery cost, etc.) and the NEV quota impact GHG emissions estimates. Detailed results and discussions of the uncertainty analyses are provided in Supplementary Note 4 and Supplementary Figs. 8–12.

**Life-cycle GHG emissions under extreme scenarios**. Two extreme scenarios that meet the CAFC standards by adopting very different vehicle technologies are created: (1) by aggressively improving ICEV efficiency without increasing BEV market share (EX-ICEV), and (2) by rapidly reaching a 50% market share of BEVs by 2030 (EX-50%BEV). In the latter scenario, it is assumed that automakers will sell less-efficient ICEVs for CAFC compliance as a result of flexibilities from excess NEV credits. Figure 1 and Table 2 show the annual GHG emissions, peak GHG emissions, and cumulative GHG emissions from 2020 to 2040 under the two extreme scenarios. The EX-ICEV scenario has significantly lower life-cycle GHG emissions compared with the EX-50%BEV scenario. Under the EX-ICEV scenario, the annual GHG emissions peak at 714 Million tonnes (Mt) in 2027, then decreases by 57 Mt to 657 Mt in 2040. Under the EX-50%BEV scenario, the annual GHG emissions continue to increase beyond 2027 and peak at 858 Mt in 2035, then decrease by 34 Mt to 824 Mt in 2040. The cumulative difference in GHG emissions from 2020 to 2040 between the two scenarios is 2339 Mt.

| Table 1 List of scenarios studied. | |
|---|---|
| **Acronym** | **Scenarios** |
| EX-ICEV | Extreme Scenario by aggressively improving ICEV efficiency |
| EX-50%BEV | Extreme Scenario by fast BEV penetration: 50% of vehicle sales by 2030 |
| DC-Reference | Dual Credit policy reference scenario |
| PT-CAFC | Policy tweak by removing NEV credit requirements in the Dual Credit policy |
| PT-BEVFC | Policy tweak by setting non-zero FC for NEVs in the CAFC standards |
| PT-M1.0 | Policy tweak by setting fuel-efficient ICEV multiplier = 1.0 in the Dual Credit policy |
| DC-HOP | Dual Credit + high oil price (HOP) |
| DC-LOP | Dual Credit + low oil price (LOP) |
| DC-LBC | Dual Credit + low battery cost (LBC) |
| DC-BCI | Dual Credit + better charging infrastructure (BCI) |
| DC-Optimistic | Dual Credit + HOP + LBC + BCI |

As discussed in the "Introduction", the treatment of BEVs under the current regulation significantly eases the difficulties in meeting CAFC standards for automakers. CAFC not only assumes the FC of BEVs to be zero, it also utilizes multipliers for each BEV produced. In the EX-50%BEV scenario, the FC of new ICEVs decreases to 6.43 L/100 km by 2025. From 2025 to 2030, rapid BEV market penetration offsets the need for additional improvements in ICEV efficiency. In the EX-ICEV scenario, however, ICEV FC decreases to 5.11 and 4.12 L/100 km by 2025 and 2030, respectively. The actual ICEV FC needed to meet the CAFC standards under the EX-ICEV and EX-50%BEV scenarios is presented and discussed in Supplementary Note 5 and Supplementary Fig. 13. Note, an ICEV FC of 4.12 L/100 km can be achieved by adopting advanced engine and hybrid technologies. For example, the Nissan Note (a compact car) and Nissan Serena (a minivan) with the e-POWER serial hybrid powertrain system, currently sold in Japan, achieved 2.7 L/100 km and 3.8 L/100 km under the JC08 driving cycle[19,20].

These two extreme scenarios clearly show that CAFC standards could result in very different GHG emissions. The primary reason is that the CAFC standards consider tailpipe GHG emissions only, ignoring the GHG emissions associated with electricity generation. This will inevitably shift the GHG emission burden to the power generation sector. Assigning zero gasoline equivalent electricity consumption to BEVs may very well entice automakers to leverage BEVs in order to meet the CAFC standards, which could impede the development of more fuel-efficient ICEV technologies.

**Impact of government policies on life-cycle GHG emissions**. The above extreme scenarios do not consider the consumer preferences on vehicle choice and the cost of implementing BEV or fuel-efficient ICEV technologies. To consider these factors, we further use the NEOCC model to quantify the technology market penetration under regulatory constraints. In addition to the Dual Credit policy, a few policy tweaks are examined to explore their potential in further mitigating GHG emissions beyond the reference case. The first tweak removes the NEV quota but maintains the CAFC requirements (PT-CAFC). The second tweak considers the electricity consumption of BEVs and PHEVs when calculating CAFC (PT-BEVFC). The Gasoline-Electricity Equivalency Factor, defined in Supplementary Note 6 and estimated in Supplementary Fig. 14, is used to convert the electricity consumption of BEVs to gasoline equivalent FC. The third tweak changes the credit multiplier for fuel-efficient ICEVs from 0.5 to

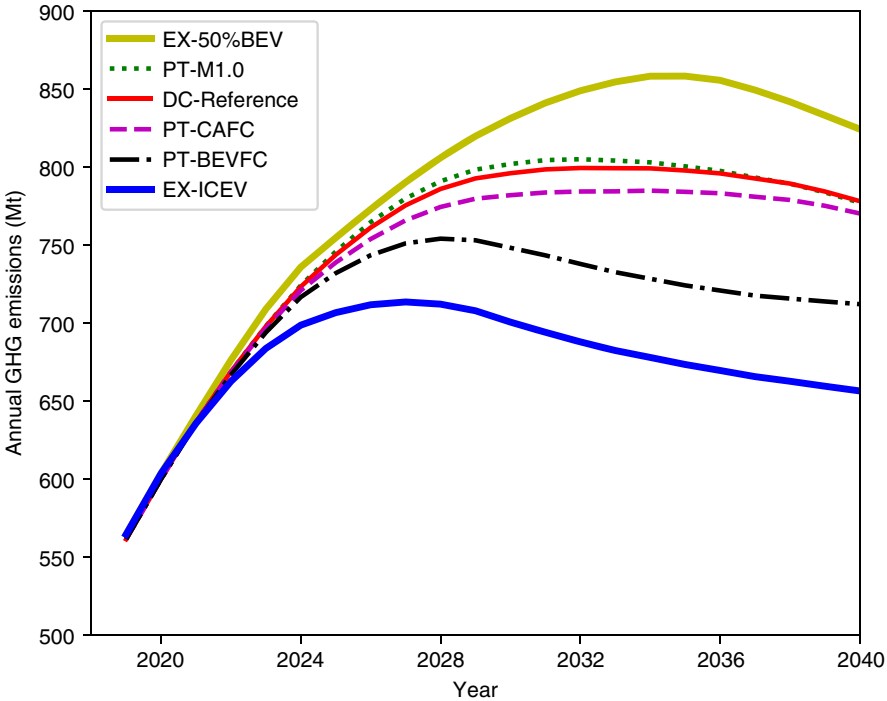

**Fig. 1 Annual life-cycle GHG emissions of the LDPV fleet in China under various policy tweak scenarios and two extreme scenarios.** EX-50%BEV: extreme scenario with 50% BEV sales share by 2030; PT-M1.0: policy tweak by setting fuel-efficient ICEV multiplier = 1.0 in the Dual Credit policy; DC-Reference: Dual Credit policy reference scenario; PT-CAFC: policy tweak by removing NEV credit requirements in the Dual Credit policy; PT-BEVFC: policy tweak by setting non-zero FC for NEVs in the CAFC standards; EX-ICEV: extreme scenario by aggressively improving ICEV efficiency.

**Table 2 Peak and cumulative GHG emissions under various policy tweak scenarios and two extreme scenarios.**

| Scenario | Year of peak GHG emissions | Peak annual GHG emissions (Mt)[a] | Cumulative GHG emissions 2020–2040 (Mt)[a] |
|---|---|---|---|
| DC-Reference | 2032 | 799 (baseline) | 15,917 (baseline) |
| EX-50%BEV | 2035 | 858 (+7.4%) | 16,606 (+4.3%) |
| EX-ICEV | 2027 | 714 (−10.6%) | 14,267 (−10.4%) |
| PT-M1.0 | 2032 | 805 (+0.8%) | 15,968 (+0.3%) |
| PT-CAFC | 2034 | 785 (−1.8%) | 15,727 (−1.2%) |
| PT-BEVFC | 2028 | 754 (−5.6%) | 15,042 (−5.5%) |

[a]The percentage in parenthesis represents the difference between that scenario and the DC-reference scenario.

1.0—this means no incentive is awarded to fuel-efficient ICEVs. Figure 1 shows the annual GHG emissions under the Dual Credit policy and policy tweak scenarios, along with the GHG emissions under the two extreme scenarios for comparison. Table 2 summarizes the peak annual GHG emissions and cumulative GHG emissions from 2020 to 2040. Figure 2 shows the GHG emissions breakdown by vehicle type, tank-to-wheel gasoline and electricity demands, new sales market share, and car stock. DC-Reference is the reference scenario.

For the DC-Reference scenario, the annual GHG emissions peak at 799 Mt in 2032 and then drop to 778 Mt by 2040, accumulating a total of 15,912 Mt from 2020 to 2040. Since the Dual Credit policy mandates NEV credits, the BEV and PHEV market shares increase steadily to 22.6% and 9.3%, respectively, by 2030. However, ICEVs still account for ~2/3 of the total vehicle stock by 2040, highlighting the importance of improving ICEV efficiency. Compared with the DC-Reference scenario, the

PT-CAFC scenario achieves lower GHG emissions, indicating that NEV credit requirements increase total GHG emissions. Under the PT-CAFC scenario, the annual GHG emissions reach a plateau at ~784 Mt between 2031 and 2036, then slowly drop to 770 Mt by 2040. Before 2026, CAFC targets are achieved by increasing the market share of both NEVs and fuel-efficient ICEVs. The market share of fuel-efficient ICEVs increases to 22.2% compared with 18.8% under the DC-Reference scenario. The above results highlight the fact that improving ICEV efficiency is a cost-effective approach to reduce total GHG emissions. The decrease in battery cost makes NEVs more competitive, and their market share steadily increases to 31.6% in 2030. During this period, the increase in NEV market share is one of the main drivers that reduces the average FC to meet the CAFC targets. After 2035, we also observe a narrowing difference in GHG emissions between the DC-Reference and PT-CAFC scenarios because the power grid becomes cleaner, causing lower life-cycle GHG emissions from NEVs. This indicates that NEVs can reduce GHG emissions in the long term. It is worth noting that the actual CAFC targets under the Dual Credit policy are slightly higher than those under the CAFC standard. This is because CAFC is a vehicle weight-based FC standard. Since BEVs are typically heavier than equivalent ICEVs[21], the higher BEV market share in the DC-Reference scenario causes a higher average vehicle weight and thus higher CAFC targets.

The multiplier for fuel-efficient ICEVs has a noticeable impact on life-cycle GHG emissions between 2025 and 2035. Compared with the DC-Reference scenario, removing the fuel-efficient ICEV multiplier (PT-M1.0 scenario) increases annual total GHG emissions by 0.6%. Beyond 2035, the impact of the fuel-efficient multiplier diminishes as BEVs become more competitive and gain a bigger market share. As a result, the Dual Credit policy imposed in this study will no longer be a constraint in technology market penetration.

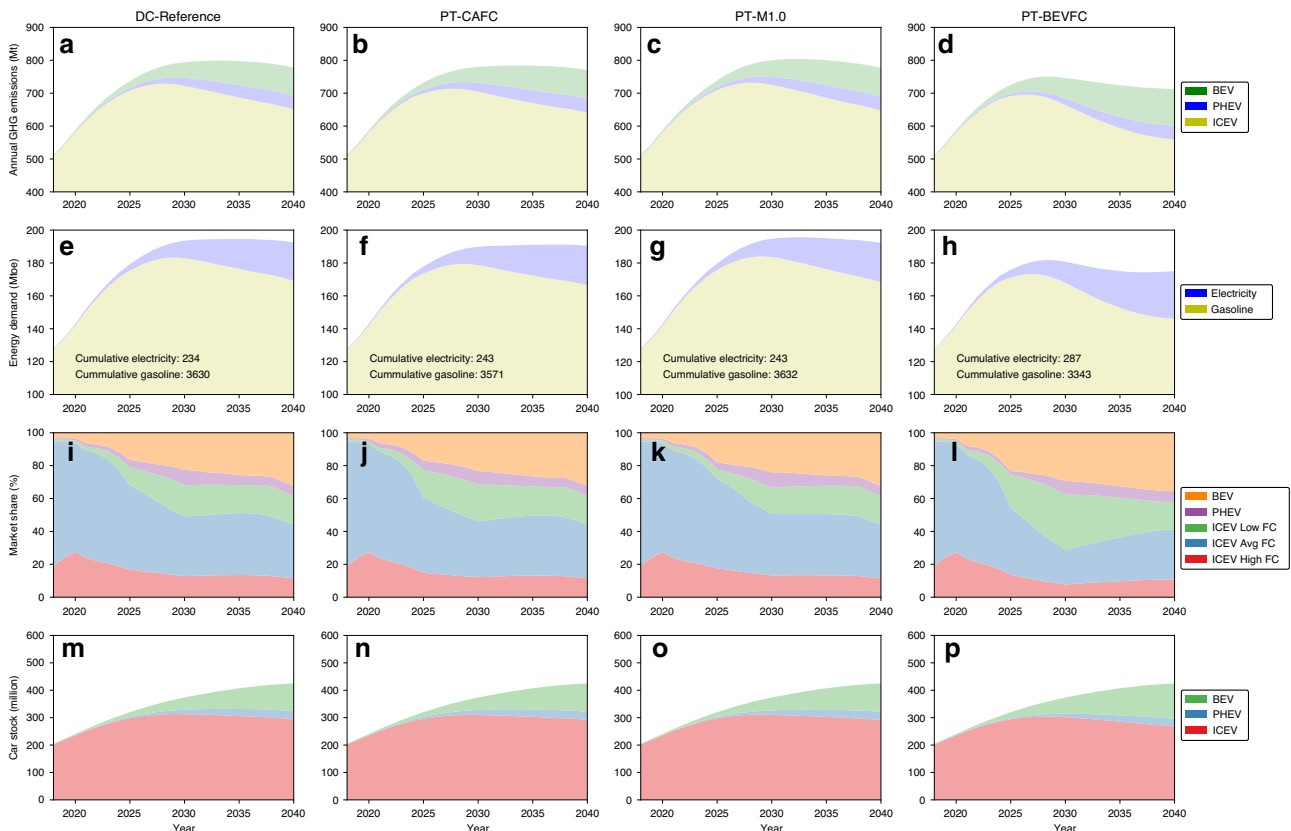

**Fig. 2 Annual GHG emissions breakdown by vehicle type, tank-to-wheel gasoline, and electricity demands, new sales market share, and car stock under various policy tweak scenarios.** Annual GHG emissions: **a** DC-Reference: Dual Credit policy reference scenario; **b** PT-CAFC: policy tweak by removing NEV credit requirements in the Dual Credit policy; **c** PT-M1.0: policy tweak by setting fuel-efficient ICEV multiplier = 1.0 in the Dual Credit policy; **d** PT-BEVFC: policy tweak by setting non-zero FC for NEVs in the CAFC standards. Annual energy demand: **e** DC-Reference; **f** PT-CAFC; **g** PT-M1.0; **h** PT-BEVFC. New vehicle sales market share: **i** DC-Reference; **j** PT-CAFC; **k** PT-M1.0; **l** PT-BEVFC. Total vehicle stock: **m** DC-Reference; **n** PT-CAFC; **o** PT-M1.0; **p** PT-BEVFC.

Significantly lower GHG emissions are observed if the electricity consumptions of BEVs and PHEVs are appropriately considered in the Dual Credit policy, as shown in the PT-BEVFC scenario. Compared with the reference Dual Credit scenario, the PT-BEVFC scenario achieves annual GHG emissions reductions of 6% and 8% by 2030 and 2040, respectively. In order to meet the CAFC targets, a sharp increase in fuel-efficient ICEVs is required —a 34% market share must be reached by 2030. Under the PT-BEVFC scenario, the average FC of ICEVs approaches 4.0 L/100 km by 2030 compared with 4.4 L/100 km by 2030 under the DC-Reference scenario. Accounting for electricity consumption of BEVs in the CAFC standards encourages auto manufacturers to achieve better fuel economy for ICEVs, which is the primary reason for the reduction in the GHG emissions.

Achieving lower GHG emissions does come with a cost, as shown in Supplementary Figs. 15 and 16. In this study, the compliance cost is the total amount of industry internal subsidies provided to NEVs and fuel-efficient ICEVs to increase their sales for policy compliance. Under all scenarios, the highest compliance cost is observed in 2030, the most challenging MY for the auto industry to meet government regulations. Beyond 2030, lower battery cost will make BEVs more cost competitive, so that it results in a greater NEV market share. In general, more-stringent policy targets can increase the cost of policy compliance. The PT-BEVFC scenario represents the most difficult regulation environment in terms of compliance, since automakers will be expected to meet the NEV credit targets and boost the sales of fuel-efficient ICEVs simultaneously. Without the credit multiplier

for fuel-efficient ICEVs, a bigger NEV market share is needed to meet the NEV credit targets. The CAFC target under the PT-CAFC scenario (only CAFC standards as the policy constraints) is also more difficult to meet compared with the Dual Credit reference scenario because no surplus NEV credits are available to compensate for the deficits of the CAFC credits.

**Factors impacting the life-cycle GHG emissions of LDPV fleet.** Five additional scenarios are carefully designed to examine the sensitivities of annual GHG emissions to externalities such as oil price, battery cost, and charging infrastructure, as illustrated in Fig. 3. Table 3 lists the peak GHG emissions and cumulative GHG emissions from 2020 to 2040 under various externalities. Additional impacts of externalities can be found in Supplementary Fig. 17 to show the GHG emissions breakdown by vehicle type, tank-to-wheel gasoline and electricity demands, new sales market share, and car stock.

Oil price is one of the most critical factors affecting total GHG emissions. Besides the oil prices used in the reference scenario (DC-Reference), two scenarios using low and high oil prices are considered. Under the low oil price scenario (DC-LOP), improving ICEV fuel economy provides fewer fuel cost savings for consumers. As a consequence, we observe a higher market share of high FC ICEVs, which causes higher average FC of new ICEVs and higher GHG emissions. Furthermore, the modeling results reveal an interesting, yet counterintuitive, observation: low oil prices could increase the NEV market share. This is because of

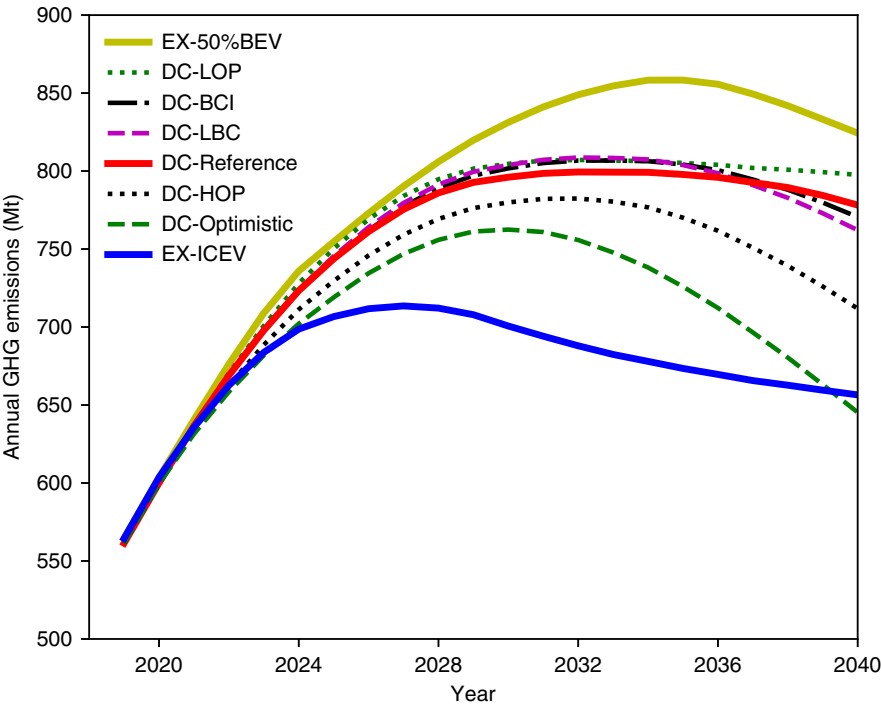

**Fig. 3 Annual life-cycle GHG emissions of LDPV fleet in China under various externalities, compared with two extreme scenarios.** EX-50%BEV: extreme scenario with 50% BEV sales share by 2030; DC-LOP: Dual Credit + low oil price; DC-BCI: Dual Credit + better charging infrastructure; DC-LBC: Dual Credit + low battery cost; DC-Reference: Dual Credit policy reference scenario; DC-HOP: Dual Credit + high oil price; DC-optimistic: Dual Credit + HOP+LBC+BCI; EX-ICEV: extreme scenario by aggressively improving ICEV efficiency.

**Table 3 Peak and cumulative GHG emissions under various externality scenarios.**

| Scenario | Year of peak GHG emissions | Peak annual GHG emissions (Mt)[a] | Cumulative GHG emissions 2020–2040 (Mt)[a] |
|---|---|---|---|
| DC-Reference | 2032 | 799 (baseline) | 15,917 (baseline) |
| DC-LOP | 2032 | 807 (+1.0%) | 16,077 (+1.0%) |
| DC-BCI | 2033 | 807 (+1.0%) | 15,959 (+0.3%) |
| DC-LBC | 2032 | 809 (+1.2%) | 15,954 (+0.2%) |
| DC-HOP | 2032 | 782 (−2.1%) | 15,437 (−3.0%) |
| DC-Optimistic | 2030 | 762 (−4.6%) | 14,880 (−6.5%) |

[a]The percentage in parentheses represents the difference between that scenario and the DC-reference scenario.

minimum NEV credit requirements and the credit multiplier (0.5) used for fuel-efficient ICEVs. Meeting NEV credit requirements reduces the market share of fuel-efficient ICEVs, which would result in more NEVs being sold to compensate for high sales of less-efficient ICEVs. Likewise, in order to meet CAFC requirements, a higher average FC for ICEVs must be offset by a higher NEV sales volume—NEVs have zero FC under CAFC guidelines. Under the DC-LOP scenario, the cumulative GHG emissions from 2020 to 2040 increase by 1.0% compared with the DC-Reference scenario. The very opposite is observed under the high oil price (DC-HOP) scenario. Under this scenario, significantly lower annual GHG emissions are observed in comparison to the DC-Reference scenario, peaking at 782 Mt (−2.1%) by 2032. By 2040, the annual GHG emissions is reduced by 8.5%, and cumulative GHG emissions drop by 3% compared to the DC-Reference scenario. It is interesting to note that neither improved charging infrastructure availability alone nor lower battery cost alone would have a significant impact on annual

GHG emissions before 2030. While both scenarios increase consumer acceptance of BEVs, they fall short of meeting the Dual Credit policy before 2030 without industry internal subsidies. These two scenarios, however, can reduce the industry's internal subsidies overall and, hence, make it easier for automakers to meet the policy requirements.

It is important to point out there are MYs that do not require the need of industry internal subsidies to meet the policy requirements (see Supplementary Figs. 15, 16). For these MYs, both CAFC targets and NEV credits are surpassed. For example, under the DC-HOP scenario, sales incentives are needed between 2023 and 2033. For other MYs, CAFC targets are surpassed, meaning the average FC of the new vehicles is lower than the CAFC target. This is the major reason that the GHG emissions in the DC-HOP scenario drop at a much faster rate than in the EX-ICEV scenario, whose GHG emissions are calculated assuming CAFC are explicitly met. In the DC-LBC scenario, the lower battery cost helps exceed the Dual Credit policy target by 2034, or 4 years earlier than the DC-Reference scenario. This is the main reason that the annual GHG emissions in the DC-LBC scenario drop at a faster rate than in the DC-Reference scenario after 2034. The DC-optimistic scenario, which combines low battery cost, better charging infrastructure availability, and high oil price, creates the most suitable environment for fuel-efficient ICEVs and NEVs. The policy requirements are surpassed for all MYs. Under the DC-optimistic scenario, annual GHG emissions peak at 762 Mt by 2030 and drop quickly to 645 Mt by 2040, lower than in the EX-ICEV scenario. The cumulative GHG emissions before 2040 are 6.5% lower than in the DC-Reference scenario. In addition, the market shares of NEV sales and stock in 2040 reach 69% and 49%, respectively.

**Impact of the GHG intensity of the electric grid.** With increasing market shares of BEVs and PHEVs, the GHG

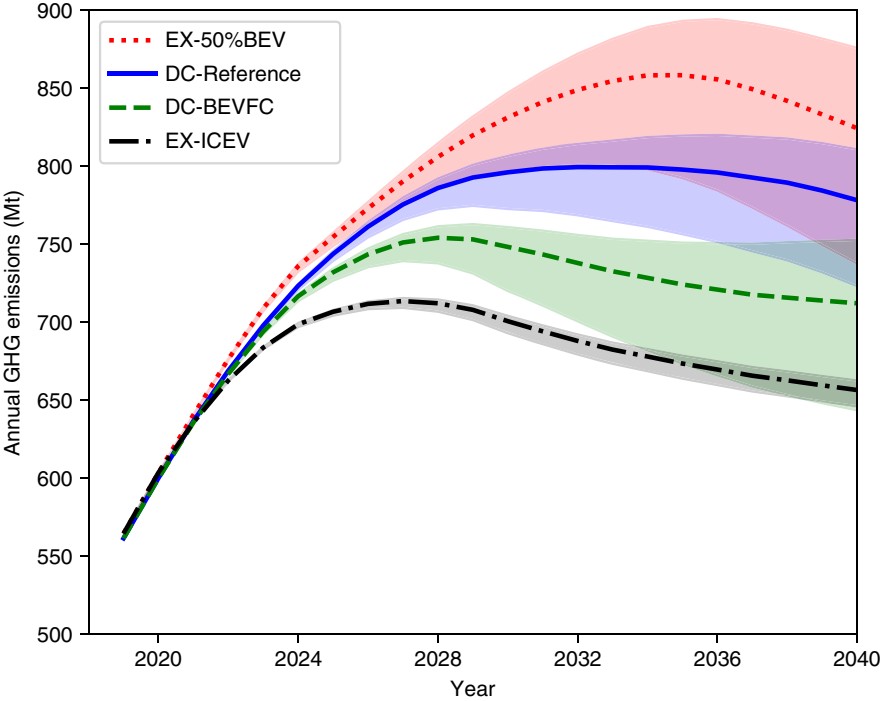

**Fig. 4 Uncertainties of annual life-cycle GHG emissions of the LDPV fleet in China due to the electricity GHG intensity.** EX-50%BEV: extreme scenario with 50% BEV sales share by 2030; DC-Reference: Dual Credit policy reference scenario; PT-BEVFC: policy tweak by setting non-zero FC for NEVs in the CAFC standards; EX-ICEV: extreme scenario by aggressively improving ICEV efficiency. The shaded area represents the uncertainties due to the GHG intensity of the Chinese electric grid.

| Table 4 Peak and Cumulative GHG emissions under three electricity GHG intensity scenarios. | | | |
|---|---|---|---|
| Scenario | Year of peak GHG emissions | Peak annual GHG emissions (Mt)[a] | Cumulative GHG emissions 2020-2040 (Mt)[a] |
| DC-Reference- RefEle | 2032 | 799 (baseline) | 15,917 (baseline) |
| DC-Reference- HighEle | 2036 | 820 (+2.6%) | 16,182 (+1.7%) |
| DC-Reference- LowEle | 2029 | 775 (−3.2%) | 15,414 (−3.2%) |
| EX-50%BEV- RefEle | 2035 | 858 (+7.4%) | 16,606 (+4.3%) |
| EX-50%BEV- HighEle | 2036 | 894 (+11.9%) | 17,021 (+6.9%) |
| EX-50%BEV- LowEle | 2032 | 801 (+0.2%) | 15,820 (−0.6%) |
| DC-BEVFC- RefEle | 2028 | 754 (−5.7%) | 15,042 (−5.5%) |
| DC-BEVFC- HighEle | 2029 | 763 (−4.6%) | 15,368 (−3.4%) |
| DC-BEVFC- LowEle | 2027 | 739 (−7.5%) | 14,423 (−9.4%) |
| EX-ICEV- RefEle | 2027 | 714 (−10.7%) | 14,267 (−10.4%) |
| EX-ICEV- HighEle | 2027 | 715 (−10.5%) | 14,334 (−9.9%) |
| EX-ICEV- LowEle | 2027 | 709 (−11.3%) | 14,137 (−11.2%) |
| [a]The percentage in parentheses represents the difference between that scenario and the DC-reference-RefEle scenario. | | | |

emissions related to electricity consumption play a more significant role. Thus, it is important to assess the uncertainties and robustness of the results owing to the changing electric grid. The Reference Technology Scenario (RTS) defined in IEA's Energy Technology Perspectives[22] is adopted as the reference (RefEle) for the changes in electricity mixes of China. IEA's RTS factors in current commitments are endorsed by the Chinese government (as well as other countries) to limit GHG emissions and improve energy efficiency. These commitments are projected to help limit the global long-term temperature rise to 4 °C. IEA also designed 6 °C and 2 °C scenarios to represent business-as-usual and highly challenging scenarios, respectively, for the global energy sector that would achieve the long-term average global temperature rises of 6 °C and 2 °C. They are used in this study to calculate the upper (HighEle) and lower (LowEle) limits of GHG intensities of

electricity. The well-to-wheels (WTW) GHG intensity of the Chinese electric grid is estimated as 185 g/MJ in 2018. With the continuous penetration of the renewable (wind, solar, etc.) and nuclear power generation technologies, the GHG intensities of electricity in 2040 are projected to be 108, 142, and 52 g/MJ for the RefEle, HighEle, and LowEle scenarios, respectively.

Figure 4 demonstrates the uncertainties due to the electricity GHG intensity for four selected scenarios. Table 4 lists the peak GHG emissions and cumulative GHG emissions from 2020 to 2040 under various scenarios in Fig. 4. The uncertainties of all other scenarios are provided in Supplementary Note 4 and Supplementary Figs. 8–12. As expected, the uncertainties increase over time owing to the higher NEV market share and the increasing differences in electricity GHG intensities between the three electric grid scenarios. A cleaner grid not only decreases

overall GHG emissions but also advances the peak GHG date. For example, under the DC-Reference-LowEle scenario, the peak annual GHG emissions occur in 2029, which is 3 years before the DC-Reference-RefEle scenario. The Ex-50%BEV scenario has the highest uncertainty because it has the highest BEV market share, whereas the EX-ICEV scenario exhibits the lowest because of low exposure to the grid. Under the LowEle scenario, the annual GHG emissions of the DC-BEVFC scenario could be lower than the EX-ICEV scenario after 2038. This clearly shows the importance of policy design and a clean electric grid in reducing GHG emissions.

## Discussion

Life-cycle GHG emissions of the Chinese passenger vehicle fleet are studied based on market penetration modeling and fleet analysis. Under the Dual Credit policy (DC-reference scenario), the total GHG emissions of the Chinese passenger vehicle fleet would not peak until 2032. However, the GHG peak would be as early as 2028 if the GHG emissions of electricity used by BEVs are properly accounted for in the Dual Credit policy.

Automakers will face increasing difficulty in meeting the Dual Credit policy before 2030. Better charging infrastructure availability and lower battery cost can mitigate the difficulties in policy compliance, but they do not have a significant impact on GHG emissions before 2035 when applied in the absence of other externalities. Beyond 2035, they have the potential to increase BEV market share and reduce GHG emissions when the electrical grid becomes cleaner. Oil price has the most prominent impact on the adoption of fuel-efficient ICEVs and BEVs. Under the Dual Credit policy, lower oil prices could increase the BEV market share and result in higher total GHG emissions.

Automotive powertrains will continue to improve technologically and increase in diversity. According to the modeling results, the NEV will increase its market share, but the ICEV will continue to dominate the passenger vehicle stock through 2040 under all credible scenarios. If ICEVs become more efficient, whether improvements are policy-driven or technology-driven, they can play a critical role in meeting the GHG regulations in China in the near and medium term. With decreasing battery cost, the BEV market share gradually increases, playing an important role in compliance with the policy mandates and reducing GHG emissions in the long term, especially as the electric grid becomes cleaner.

Based on our analyses, several recommended adjustments (tweaks) to the Dual Credit policy would help China to meet its future climate targets. First, transportation policies should consider emissions based on the whole life cycle. The electricity-based GHG emissions of BEVs and PHEVs should be accounted for in the calculations. As shown, this can significantly reduce the emissions leakage effect without affecting BEV sales. With an increasing NEV market share in the future, a cleaner electric grid will play a more significant role in reducing the life-cycle GHG emissions of the transportation sector. Thus, in the long term, transportation policies should be coupled with other policies to accelerate the decarbonization of the electric grid by offering stakeholders some regulatory incentives to reduce grid emissions. Second, policy design should be based on performance rather than technology. As shown in this study, removing technology-based NEV mandates from the Dual Credit policy would achieve lower GHG emissions. Third, the actual fuel consumption level of ICEVs should be monitored annually to ensure that new ICEV efficiency continues to improve regardless of the NEV credits available for compliance with policy mandates.

## Methods

In this study, two quantitative models—the NEOCC model and China Vehicle Fleet model are used. Figure 5 shows the high-level model diagram describing the interaction between the two models, as well as parameter inputs and the outputs. The methodologies used in the models are described in this section, and more detailed information is provided in Supplementary Notes 1 and 2, Supplementary Figs. 3–7, and Supplementary Table 3.

**NEOCC model.** The NEOCC, developed by the Oak Ridge National Laboratory, is calibrated with 2016–2019 Chinese passenger vehicle market data and is used for modeling vehicle industry compliance and market sales dynamics. The model assumes the vehicle industry will attempt to maximize industry profit by distributing its internal subsidies for different vehicle types, subject to the policy compliance constraints. The genetic algorithm is adopted to determine the optimal allocation of internal subsidies for obtaining the highest industry profits. The NEOCC model divides vehicles into three categories: ICEV (including both conventional fossil fuel-powered models and hybrid models), BEV and PHEV. A total of 16 different vehicle types are included in the NEOCC version released in 2020. Conventional ICEVs include three types: high, medium, and low FC vehicles. For BEV sedans, five BEVs with different electric driving ranges are considered: 150 km, 200 km, 250 km, 300 km, and 400 km. For BEV SUVs/crossovers, two BEVs are considered: 250 km, and 350 km.

The NEOCC model is an optimization model combined with a discrete choice model that specifies the probabilities of the consumers (individual and fleet vehicle buyers) choosing alternative vehicle types. The probabilities of the selections are calculated based on the utility function, which considers multiple variables such as vehicle prices, fuel/electricity consumption cost, government subsidies, industry internal subsidies, charging inconvenience cost, and constants. These variables are calibrated using historical vehicle market data. In determining vehicle type selection, the model assumes the auto industry will maximize its total profit value by allocating the variable—internal subsidies for a series of different vehicle technologies under the policy constraints, as shown in Supplementary Note 2. Iterating with the genetic algorithm, the market dynamics can ultimately achieve an equilibrium where the industry achieves the maximum profit under the policy constraints[1]. The objective function is shown in Eq. (1). The constraints of the Dual Credit policy include the NEV credit rules, as shown in Eq. (2), the CAFC credit rules, as shown in Eq. (3), and the combination of NEV credit rules and CAFC credit rules, as shown in Eq. (4).

$$\text{Max}_x R(x) = P(x) - I(x) - M(x) \tag{1}$$

$$s.t. \, N(x) \geq 0 \tag{2}$$

$$s.t. \, C(x) \geq 0 \tag{3}$$

$$s.t. \, N(x) + C(x) \geq 0 \tag{4}$$

where, $R(x)$ in Eq. (1) is the industry's total gross profit to be optimized and $x = [\chi_1, \ldots, \chi_t, \ldots, \chi_T]$ is an array that groups a set of decision variables (industry's internal subsidies) to realize this optimization. $\chi_t$ describes the industry's internal subsidies for the vehicle type $t$. $P(x)$ is the sum of the products of vehicle prices by vehicle type and the corresponding sales volumes by vehicle type. $I(x)$ is the sum of the products of industrial internal subsidies by vehicle type and the corresponding sales volumes. $M(x)$ is the sum of the products of production cost by vehicle type and the corresponding sales volumes. $C(x)$ is the sum of the products of CAFC credits achieved by vehicle type and the corresponding sales volumes. $N(x)$ is the sum of the products of NEV credits achieved by vehicle type and the corresponding sales volumes. The calculations of CAFC credits and NEV credits for different vehicle types are explained by the Dual Credit policy[5]. The sales volumes for these vehicle types are calculated through the discrete choice method in the NEOOC model. The algorithm logic flow is explained in Supplementary Fig. 3. More detailed information is provided in studies by Ou et al.[17,23,24]. The simulated market share of different vehicle technologies and their corresponding fuel consumption rates (FCRs from the NEOCC model are then used as inputs to the China Vehicle Fleet model to calculate the GHG emissions of the LDPV fleet in China through 2040.

**China Vehicle Fleet model.** The China Vehicle Fleet (China-Fleet) model developed by Argonne National Laboratory (ANL)[25,26] is used to estimate the energy consumption and GHG emissions of the LDPV fleet in China on a life-cycle basis. The model includes historical and projected data of detailed vehicle types/technologies (vehicle stocks, sales, market shares, fleet turnovers, mileage traveled, fuel economy, etc.) and estimates the WTW energy consumption and GHG emissions of individual vehicle types/technologies based on the upstream intensities of different transportation fuels derived from the ANL's GREET (Greenhouse Gases, Regulated Emissions, and Energy Use in Transportation) model[27].

The WTW analysis accounts for both the well-to-tank (WTT) and the tank-to-wheels (TTW) stages[25,27,28]. WTT includes processes related to the production and distribution of fuels prior to fueling or charging, and TTW covers consumption and emissions from vehicle operation. The TTW energy consumption of the

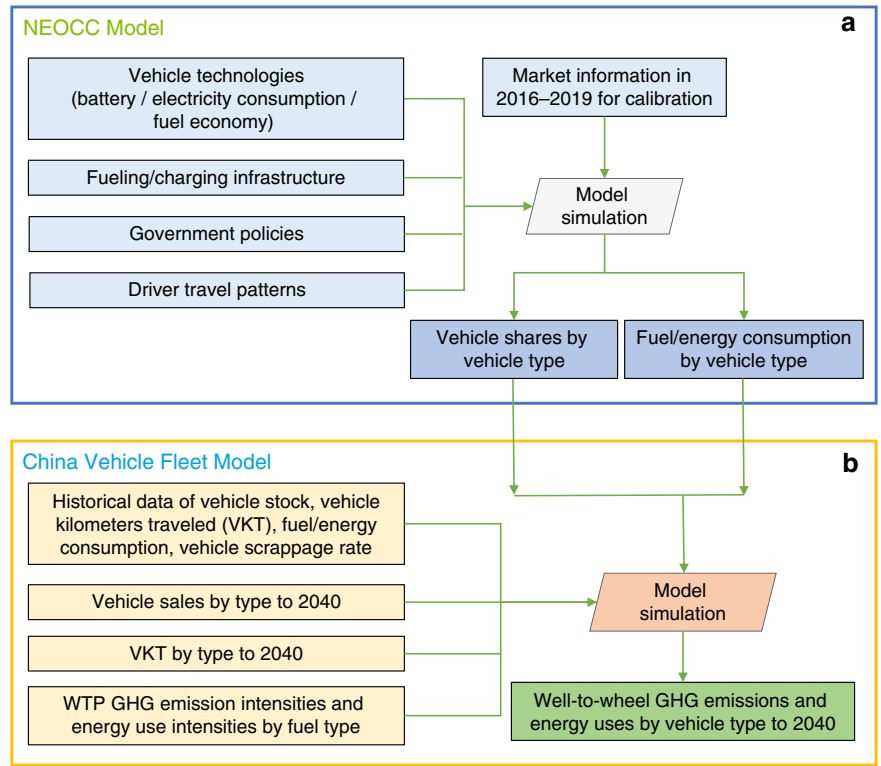

**Fig. 5 High-level model diagram with key inputs and outputs. a** The data flow diagram of the NEOCC Model. The rectangle boxes in light blue show the key inputs of the NEOCC model. The parallelogram box in gray shows the NEOCC model simulation. The rectangle boxes in dark blue show the outputs from the NEOCC model. **b** The data flow diagram of the China Vehicle Fleet Model. Rectangle boxes colored in yellow show the key inputs of the China Vehicle Fleet Model. The NEOCC Model outputs colored in dark blue of **a** are also used as the inputs of the China Vehicle Fleet Model. The parallelogram box in orange shows the China Vehicle Fleet Model simulation. The rectangle box colored in green shows the final output of the model analysis.

vehicle fleet in year $m$ is determined by the number of vehicles (i.e., stock), VKT, and vehicle FCR in J/km in Eq. (5)[25]:

$$Energy_{TTW,m} = \sum_k Energy_{TTW,k,m} = \sum_k \sum_i \sum_j (Stock_{i,k,j,m} \cdot VKT_{i,k,j,m} \cdot FCR_{i,k,j,m}) \quad (5)$$

where $k$, $i$, and $j$ represent fuel type, vehicle technology, and vehicle age, respectively. The vehicle stocks, VKT, and historical FCRs are estimated based on the methods and parameters in the base case of the China Vehicle Fleet model[25] with updates based on a recent study of private LDPV sales and stocks[26]. In the present study, we recognize that there are gaps between the vehicle labeled and real-world FCRs and therefore include the labeled-to-real-world adjustment in the China-Fleet model to reflect the corresponding impacts. Based on previous research results, we adjust the labeled FCRs/ECRs to the real-world ones by multiplying a ratio of 1.2 for ICEVs[25,29] and 1.4 for BEVs[28,30,31]. Updates and adjustments to the base case of the China Vehicle Fleet model for this study can be found in Supplementary Table 4. The future FCRs and market shares of individual vehicle technologies are derived from scenario analysis of extreme BEV and ICEV market penetration and modeling results by the NEOCC model.

WTW energy consumption and GHG emissions in year $m$ are further determined by the TTW energy consumption of different fuel types ($k$) and their corresponding WTW energy consumption intensities ($EI_{WTW,k,m}$) and GHG emissions intensities ($GI_{WTW,k,m}$) as follows:

$$Energy_{WTW,m} = \sum_k Energy_{WTW,k,m} = \sum_k (Energy_{TTW,k,m} \cdot EI_{WTW,k,m}) \quad (6)$$

$$GHG_{WTW,m} = \sum_k GHG_{WTW,k,m} = \sum_k (Energy_{TTW,k,m} \cdot GI_{WTW,k,m}) \quad (7)$$

$EI_{WTW,k}$ and $GI_{WTW,k}$ represent the associated amount of energy consumed and GHG emitted from all processes for one energy unit of fuel $k$ in both the WTT and TTW stages[25,27].

The GHG intensities of liquid fuels and electricity generation are needed to calculate the GHG emissions of the whole fleet. The WTW GHG emissions intensity of gasoline is assumed to decrease from 88.2 g/MJ in 2018 to 86.5 g/MJ in 2040 owing to advancing technology in oil extraction and refining[25,32]. The historical WTW GHG emission intensities of electricity are estimated based on the electricity mixes by generation type (coal, oil, natural gas, etc.), their corresponding

generation efficiencies, and transmission losses reported by IEA[33], National Bureau of Statistics of China (NBSC)[34], and China Electricity Council[35].

**Reporting summary.** Further information on research design is available in the Nature Research Reporting Summary linked to this article.

## Data availability

All data regarding the parameters used in the NEOCC model and China-Fleet Vehicle model and data sources are documented in Supplementary Information. More specifically, the data used for NEOCC model calibration are presented in Supplementary Table 3. All other data (gasoline prices, battery cost, charging infrastructure availability, home charging availability, public charging availability) used in the study are given in Supplementary Figs. 4–7. The details of NEV rules adopted in the NEOCC model is presented in Supplementary Tables 1 and 2. The sales, stocks, and vehicle miles traveled used in the Base Case of China Vehicle Fleet Model are presented in Supplementary Table 4. Source data are provided with this paper.

## Code availability

The Microsoft-Excel-based NEOCC model and China Vehicle Fleet model are available from the corresponding authors upon request.

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

## Acknowledgements

This research was financially supported by Aramco Services Company and used resources at the National Transportation Research Center at Oak Ridge National Laboratory and Systems Assessment Center of Energy Systems Division at Argonne National Laboratory. This manuscript has been authored by UT-Battelle, LLC under contract no. DE-AC05-00OR22725 and UChicago Argonne, LLC under Contract DE-AC02-06CH11357 with the US Department of Energy. The authors are solely responsible for the contents of the paper.

## Author contributions

X. He, S. Ou, Y. Gan, and Z. Lu conceived of and designed the study. S. Ou, Y. Gan, X. He, and Z. Lu involved in data gathering, processing, and analysis. S. Ou, X. He, Z. Lin, and R. Yu contributed to the NEOCC model development. Z. Lu, Y. Gan, X. He, Y. Zhou, M. Wang, S. Przesmitzki, and J. Bouchard contributed the China-Fleet Model development. The results were interpreted by X. He, S. Ou, and Y. Gan with critical input from A. Amer and L. Sui. The writing of the paper was led by X. He and S. Ou and all co-authors contributed to the review and revision.

## Competing interests

The authors declare no competing interests.
