## [Peer Review File · Nature Communications]

REVIEWER COMMENTS

Reviewer #2 (Remarks to the Author):

The paper "Greenhouse Gas Consequences of the China Dual-Credit Mandate" by He et al. is a novel examination of the emissions impacts of passenger vehicle policy in China. The topic itself is a timely, as the landscape of policy intervention and regulations for both fuel consumption rates and electrification of vehicles in China is rapidly evolving. The authors conduct a fairly broad set of policy analyses scenarios including extreme market penetration, business-as-usual, policy "tweaks" (affecting credit structure in the CAFC and NEV policies), and a range of macroeconomic factors.

However, there are several major concerns that the authors must address before publication should be considered:

1. A serious English language editor should be employed, the language is very irregular and contains many grammatical mistakes. The deficiencies are nearly serious enough to make portions of the text difficult to comprehend. Extensive use of passive voice.
2. Significantly more detail on the regulatory policies are needed to describe both the Chinese CAFC and NEV policies. Non-expert readers (of which there will be many in a broad Nature journal) will have a difficult time understanding the nuances of the credit system—especially as they remain essentially undefined in the text. Some historical context is needed, as well as a timeline of the requirements of the regulation (fuel consumption in CAFC and ZEV requirement in NEV), and a clearly outlined reasoning for assumptions about the future CAFC/NEV rules. The authors place most of this detail in the Supplemental Information, but given its importance to the main text, this material should most definitely be moved into the main text (or at least partially).
3. The methods surrounding the modeling in this paper are not very clear. While some details are provided on how each of the modules (NEOCC and the China Vehicle Fleet Model) work, some framework is needed to describe the overall approach that the authors took in getting from some set of inputs to their final outputs. Perhaps some high-level model diagram describing the interaction between the modules, as well as parameter inputs and the outputs. Research in high impact journals ought to strive to carry the burden of reproducibility, but it would be very difficult to reproduce the work conducted here due to the ambiguity in the methods.
4. The robustness of the results of this study are unknown. It would behoove the authors to prioritize conducting uncertainty analysis to understand how important certain assumptions are in their final results (e.g. electricity grid carbon intensity, battery costs, etc.).

More detailed comments:

Paragraph starting at line 50, needs to expand on what the "Dual Credit" policy is. There is no explanation of credits for either CAFC or NEV, instead the authors dive straight into the Dual Credit discussion without explanation.

Line 62, what is the petroleum equivalency factor?

Lines 73-74, authors presume that the readers will know what the CA ZEV mandate is without explanation.

Line 132, "Super credits" are not explained

Figure 1, should consider starting y-axis at 0 rather than 500

Lines 175-178, this is an important point and should be expanded on. What would the cost counterfactual be in the long-term between scenarios?

Figure 2, generally in the text as well, the authors should consider displaying the cumulative CO₂ across the scenarios

Lines 231-232, the authors pose that higher average FC ICVs "commands" higher NEV sales volume—there is an inherent assumption throughout the results that indicates that ICV FC will be tied to NEV sales. It is important to note that this is only true if the CAFC regulation is a binding constraint—if automakers decide to violate the rule and pay a fine (unlikely) or if they exceed the credit requirements (more likely in scenarios with high NEV), there is no "guarantee" that they will automatically make the vehicles in the rest of their fleet "dirtier". At the very least, this assumption should be noted in the text.

Line 285, "transportation policies should consider emissions based on the whole life cycle"—why?

If NEVs/ZEVs' upstream emissions must be accounted for by automakers, with what mechanism do these companies have to improve the grid emissions? Perhaps it would be better to state that transportation policies should be coupled with policies that clean the grid, giving the correct stakeholders some regulatory incentive to improve grid emissions (rather than OEMs which have no power in this domain).

Line 294, "...calibrated with 2017 Chinese passenger vehicle market data...", the vehicle choice model being calibrated to a single year appears to be relatively problematic. How do the authors account for changes in consumer preferences over time?

Equation 1, why are profits being minimized? Shouldn't profits be maximized (or I suspect the authors were trying to minimize the negative profit).

Equations 2-4, these equations are nonsensical. $C(x)$ and $N(x)$ and not adequately explained. The equations read: "CAFC credits and NEV credits should be greater than or equal to 0".

Lines 358-368, considerably more effort should be placed in the forecasts of the electricity grid. It is unclear how robust the results are to the changing electricity grid, but at a minimum these should be tested at the same level of robustness/detail as scenarios of CAFC and NEV given how likely the grid scenarios will be critical to the final results of the paper.

Reviewer #3 (Remarks to the Author):

This paper addresses a timely policy issue that is-- the linkage and interactions between China's CAFC and NEV Mandate could derail the efforts to improve ICEVs efficiency and therefore increase GHG emissions. By using different scenarios, especially the two extreme ones, the authors highlighted the severity of the drawbacks in the designing of China's Dual-Credit Policy.

The analysis is solid and scenarios are carefully designed. The two models undergird this paper are well-developed and well-known.

I have three issues with this paper, but they in no way affect the overall efficacy of this paper.

1) The assumption that Chinese cars' real CAFC compliance. As demonstrated in the VW Diesel-gate and many of the subsequent studies, new car CAFC compliance could be problematic, China included. On top of that, as auto components deteriorate, the gap between the CAFC target and the real FC gets bigger. One of the arguments for promoting BEVs is that relatively speaking BEVs tend to have long-life reliability in terms of energy consumption. So there could be a 10-20% gap between the real world CAFC and the "paper" CAFC readings in car's life time. I hope the authors could address this issue.

2) In assuming NEV credit requirement. The authors correctly mentioned that the credit requirement in 2023 is 18%, however, the authors then assumed that in 2030, the NEV credit requirement would be 40% (supposedly to match the 40% NEV market-share target). This is wrong. Assuming the average range of all the NEVs in 2023 is 200 km, then meeting 18% of NEV requirement would only need 11.3% of NEVs in total market share in 2023. If we assume the same conversion rate, then to reach 40-50% market share in 2030, China's NEV credit requirement would need to raise to 64-80% of NEVs with the same 200 km BEVs (and PHEVs). In short, NEV Credit requirement is larger in percentage than in vehicle numbers.

3) Line 168, "the BEV and PHEV market shares increase steadily to 21.5% and 14.2% by 2030." PHEVs share assumed here is too big. In 2019, the share of PHEVs in total PEVs was only 21%. Globally, PHEVs are out of fashion. GM stopped selling Volt. A reasonable assumption would be 5-10% of the total PEVs in 2030.

In conclusion, this is a very good paper. I would love to hear the authors' responses.

Response Letter to Reviewer Comments

The authors would like to thank the efforts of the editorial personnel and the reviewers, especially during this pandemic period. Comments are in black and numbered. Our responses are in blue, and quoted texts from the manuscript are in purple and italic.

Reviewer #2:

The paper “Greenhouse Gas Consequences of the China Dual-Credit Mandate” by He et al. is a novel examination of the emissions impacts of passenger vehicle policy in China. The topic itself is a timely, as the landscape of policy intervention and regulations for both fuel consumption rates and electrification of vehicles in China is rapidly evolving. The authors conduct a fairly broad set of policy analyses scenarios including extreme market penetration, business-as-usual, policy “tweaks” (affecting credit structure in the CAFC and NEV policies), and a range of macroeconomic factors. However, there are several major concerns that the authors must address before publication should be considered:

The authors really appreciate the invaluable comments and suggestions for this paper.

Reducing GHG emissions in the transportation sector is crucial for the Chinese government to accomplish its commitment of peaking CO₂ emissions around 2030. The “Measures for Passenger Cars CAFC and NEV Credit Regulation” policy, or more widely known as the Dual Credit policy, is the major policy that the Chinese government enacted to reduce vehicle fuel consumption and promote BEV market penetration. However, the regulation considers the vehicle tail-pipe emissions only; ignoring the emissions during fuel and electricity generation may cause unintended consequences. The main purpose of the paper is to assess the GHG emissions from the well-to-wheels perspective. The paper also considers constraints from the vehicle policies and vehicle market dynamics; the latter also takes into account technology evolution and consumer choices.

As far as we know, the NEOCC is the first model that considers consumer choice, government policy, vehicle technology cost (both production and in-use), fueling/charging infrastructure, and industry choice to holistically assess the future China’s vehicle market share. When combined with the China Vehicle Fleet model, we are able to comprehensively assess the GHG emissions under various policy and electricity carbon intensity scenarios.

Since the first submission of the paper to Nature Communications in February, we have been working on model refinement with the China Automotive Technology and Research Center (CATARC), a leading Chinese organization that developed the Dual Credit policy. We also took your suggestions into considerations on the model revisions. Below are the major updates we made to the NEOCC model for this paper:

- (1) Recalibrated the NEOCC model with 2016-2019 Chinese passenger vehicle market data.
- (2) Considered full-hybrid ICEVs only as the “ICEV Low FC” technology which qualifies for the “Low Fuel Consumption” credit in the NEOCC model. The major consequence is the reduction of market shares in “ICEV Low FC”. The micro/mild-hybrid ICEVs are included

into the “ICEV Avg FC” technology, which helps reduce the FC of “ICEV Avg” at a faster rate. The classifications of “ICEV Low FC” and “ICEV Avg FC” are given by Supplementary Table S4.

- (3) Updated the vehicle production cost based on the 2016-2019 vehicle sales and MSRP data provided by CATARC. The paper discussed the detailed cost analysis of the Chinese vehicle market that was just published by Energies [1].
- (4) Reduced the reference battery production cost from \$116/kWh to \$104/kWh by 2030 to reflect the recent battery cost reduction trend in China. Overall, BEVs become more competitive in China.
- (5) Considered the efficiency improvement of BEVs, which reduces the battery size required for a specific driving range.
- (6) Reduced the projection of gasoline price in 2020 to reflect the COVID-19 impact.

Some simulation results shown in this revised manuscript are slightly changed as the model and assumption were revised. However, these variations do not change the general conclusions of the paper. Following summarizes the two major changes:

1. The market share of “ICEV Low FC” technology in the updated model version is smaller than it was calculated in the older model version, because the “ICEV Low FC” technology is calibrated to full-hybrid ICEVs only in the new model version.
2. In the new model version, the NEV market share shows a gradual increase starting 2020 under the “PT-CAFC” scenario. However, in the older model version, the market share of NEVs maintains at about 8% before 2026, and then quickly increases to 24% by 2030. This change is mainly due to the item (2) mentioned above. Setting a more stringent “ICEV Low FC” requirement increases the production volumes of “ICEV Low FC”. Meanwhile, the lower battery cost assumption (item (4)) also improves the competitiveness of NEVs. As a consequence, we observe a smoother increase in the NEV market share.

It is true that the NEOCC model and the China Vehicle Fleet model are not easy to reproduce as each model takes more than two years to develop. The China Vehicle Fleet Model developed by Argonne National Laboratory has been released to some prestigious research organizations, such as SAE-China, U.S. EIA, IEA, MIT, ICCT, Energy Innovation, International Transport Forum. The NEOCC model developed by Oak Ridge National Laboratory has been released to CATARC for future policy development and feasibility assessment. Both models will be submitted along with the revised paper for your review. Upon the publication of this paper, we will make these models available to the researchers/organizations who are interested in working with us on relevant research topics.

Reference:

[1] Ou, S.; Li, W.; Li, J.; Lin, Z.; He, X.; Bouchard, J.; Przesmitzki, S. Relationships between Vehicle Pricing and Features: Data Driven Analysis of the Chinese Vehicle Market. *Energies* 2020, 13, 3088.

1. A serious English language editor should be employed, the language is very irregular and contains many grammatical mistakes. The deficiencies are nearly serious enough to make portions of the text difficult to comprehend. Extensive use of passive voice.

Thanks for your comment. We extensively reviewed the paper to make sure the language is consistent and to avoid grammatical errors. Since the first draft, we have used a professional editor and several native English speakers to refine the paper. We expect this version's grammar is of higher quality and will note this for future submission.

2. Significantly more detail on the regulatory policies are needed to describe both the Chinese CAFC and NEV policies. Non-expert readers (of which there will be many in a broad Nature journal) will have a difficult time understanding the nuances of the credit system—especially as they remain essentially undefined in the text. Some historical context is needed, as well as a timeline of the requirements of the regulation (fuel consumption in CAFC and ZEV requirement in NEV), and a clearly outlined reasoning for assumptions about the future CAFC/NEV rules. The authors place most of this detail in the Supplemental Information, but given its importance to the main text, this material should most definitely be moved into the main text (or at least partially).

Thanks for the excellent suggestions. We completely agree that it is important to provide readers enough information about the Chinese CAFC and NEV policies in the main text. We have added additional information to more adequately describe the Dual Credit policy in the main text, and we added more detailed explanations in the Supplementary Material S1. However, due to the word limit Nature Communications places on research papers, we were somewhat limited in the amount of detail we could provide in the main text and hope our revisions are adequate. To augment the Supplementary Material, we added a timeline of the requirements of the regulations and described the reasons for assumptions about the future CAFC/NEV rules. In addition, based on your suggestions below, we conducted an uncertainty analysis for the NEV quota by varying the 2030 NEV quota target from 30% to 50% (Supplementary Material S8.1).

Figure 1. Timeline of the existing Dual Credit policy and the future assumptions used in the NEOCC model

The following text are added in the main text:

“The new Dual Credit policy consists of two components: (1) CAFC credit rules and (2) NEV credit rules. First, the CAFC credit rules set targets for the production-weighted average FC for vehicle manufacturers. Adjusting for projected sales and vehicle curb weight, the projected CAFC target by 2025 would be about 4.7 L/100 km. Second, the NEV credit rules mandate that manufacturers produce enough NEVs to meet the NEV credit quota. This quota starts at 10% in 2019 and increases by 2% per annum until 2023. To qualify for NEV credit, the vehicle must be a battery electric vehicle (BEV), plug-in hybrid electric vehicle (PHEV), or fuel cell vehicle (FCV). The number of NEV credits granted to each NEV will also vary depending on vehicle powertrain type, vehicle weight, electric driving range, and vehicle energy efficiency. Details of the vehicle policies are provided in Supplementary Material S1. It is believed that the Dual Credit policy will become the driving force for NEV growth in the near future⁴.”

3. The methods surrounding the modeling in this paper are not very clear. While some details are provided on how each of the modules (NEOCC and the China Vehicle Fleet Model) work, some framework is needed to describe the overall approach that the authors took in getting from some set of inputs to their final outputs. Perhaps some high-level model diagram describing the interaction between the modules, as well as parameter inputs and the outputs. Research in high impact journals ought to strive to carry the burden of reproducibility, but it would be very difficult to reproduce the work conducted here due to the ambiguity in the methods.

Thanks for the excellent suggestion. In the “Methods” section, we added the figure below to describe the interaction between these two models, as well as key parameters and inputs and outputs of the models.

We agree that it would be very difficult for an outsider to easily reproduce the NEOCC model and/or the China Vehicle Fleet model, as each model took more than two years to develop. Upon the publication of this paper, we will make these models available to the researchers/organizations who are interested in working with us on relevant research topics.

Figure 2. High-level model diagram with key inputs and outputs

4. The robustness of the results of this study are unknown. It would behoove the authors to prioritize conducting uncertainty analysis to understand how important certain assumptions are in their final results (e.g. electricity grid carbon intensity, battery costs, etc.).

Thank you for the excellent suggestion to improve the paper. We added a dedicated section in Supplementary Material S8 describing the uncertainties of key model assumptions.

In the uncertainty analysis, we considered the following NEOCC model assumptions: battery cost, oil price, charging infrastructure availability, and NEV quota. We also quantified the uncertainties due to the GHG intensity of the Chinese electric grid for each scenario. As an example, the figure below shows the uncertainties due to the battery cost. The shaded area represents the uncertainties due to the GHG intensity of the Chinese electric grid. Please see Supplementary Material S8 for details on other assumptions/scenarios.

Furthermore, we summarized all the results generated for this paper in a tabular form, and organized it so that readers can do the comparison themselves.

Figure 3. Uncertainties of GHG emissions due to battery cost. The shaded area represents the uncertainties due to the GHG intensity of the Chinese electric grid.

More detailed comments:

Paragraph starting at line 50, needs to expand on what the “Dual Credit” policy is. There is no explanation of credits for either CAFC or NEV, instead the authors dive straight into the Dual Credit discussion without explanation.

Thank you for the suggestion to improve the paper. We agree that it is important to briefly introduce the Dual Credit policy in the main text, as this is the key policy assessed in this work. The following text has been added in the main text to provide some information on the policy:

“The new Dual Credit policy consists of two components: (1) CAFC credit rules and (2) NEV credit rules. First, the CAFC credit rules set targets for the production-weighted average FC for vehicle manufacturers. Adjusting for projected sales and vehicle curb weight, the projected CAFC target by 2025 would be about 4.7 L/100 km. Second, the NEV credit rules mandate that manufacturers produce enough NEVs to meet the NEV credit quota. This quota starts at 10% in 2019 and increases by 2% per annum until 2023. To qualify for NEV credit, the vehicle must be a battery electric vehicle (BEV), plug-in hybrid electric vehicle (PHEV), or fuel cell vehicle (FCV). The number of NEV credits granted to each NEV will also vary depending on vehicle powertrain type, vehicle weight, electric driving range, and vehicle energy efficiency. Details of the vehicle policies are provided in Supplementary Material S1. It is believed that the Dual Credit policy will become the driving force for NEV growth in the near future ⁵”

Line 62, what is the petroleum equivalency factor?

Thank you for pointing out that we did not describe this adequately. We modified the sentence related to petroleum equivalency factor to address this:

“In the U.S., the Corporate Average Fuel Economy (CAFE) standard calculates the petroleum equivalent fuel economy of BEVs using the petroleum equivalency factor (PEF). For BEVs without petroleum-powered accessories the PEF is 33,705 Wh/gal ⁶. This PEF considers the upstream efficiency of fuel production, the national average electricity generation, transmission efficiencies, and uses an incentive factor of 0.15 to further stimulate auto manufacturers to produce BEVs.”

Lines 73-74, authors presume that the readers will know what the CA ZEV mandate is without explanation.

Thanks for the comment. We agree we did not adequately describe the CA ZEV mandate. We changed the text to clarify the policy to:

“In the U.S., the California state government’s Zero Emission Vehicle (ZEV) regulation requires automakers to sell BEVs, PHEVs, or FCVs to meet ZEV credit mandates. According to ZEV policy, multipliers are applied when counting certain ZEV sales.”

Line 132, “Super credits” are not explained

Thank you for the comment. “Super credits” is a nickname for credit multipliers for certain NEVs. They are an incentive for automakers to help increase NEV production and sales. For example,

in 2016, the multiplier can be up to 5 for some NEVs. It is called “super” because the multiplier significantly eases the difficulty of achieving the CAFC target. In the revised paper, we removed the “super credits” expression in the main text for clarity because it is unnecessary.

Figure 1, should consider starting y-axis at 0 rather than 500

Thank you for the suggestion. The figures below show the graph with two different y-axes, one at 0 and one at 500. Both have advantages. Setting the y-axis at 0 better shows the cumulative GHG emissions. We prefer keeping the y-axis at 500 because it better uses the space comparing the difference between scenarios and better shows the difference between the different scenarios.

Lines 175-178, this is an important point and should be expanded on. What would the cost counterfactual be in the long-term between scenarios?

Thank you for the comment and question.

In the new version of the model, the NEV market share shows a gradual increase starting in 2020 under the “PT-CAFC” scenario. In the older version of the model, however, the market share of NEVs stays at about 8% until 2026, and it does not increase to 24% until 2030. This is mainly because only full-hybrid ICEVs are considered as the ICEV-Low-FC technology that qualifies for the Low Fuel Consumption credit in the new version of the model. Setting a more stringent fuel consumption requirement in this scenario increases the production volumes of “ICEV FC”. Meanwhile, the lower battery cost assumption in the new model also improves the competitiveness of NEVs. The new model shows that automakers should continue advancing ICEV efficiency while promoting BEV market share because this is the most cost-effective approach to meet the future regulatory requirements.

Based on the new results, we changed the text to:

“Under the PT-CAFC scenario, the annual GHG emissions reach a plateau at about 784 Mt between 2031 and 2036, then slowly drop to 770 Mt by 2040. Before 2026, CAFC targets are achieved by increasing the market share of both NEVs and fuel-efficient ICEVs. The market share of fuel-efficient ICEVs increases to 22.2% compared with 18.8% under the DC-Reference scenario. The above results highlight the fact that improving ICEV efficiency is a cost-effective approach to reducing total GHG emissions. The decrease in battery cost makes NEVs more competitive, and their market share steadily increases to 31.6% in 2030. During this period, the increase in NEV market share is one of the main drivers that reduces the average FC to meet the CAFC targets.”

Figure 2, generally in the text as well, the authors should consider displaying the cumulative CO2 across the scenarios

Thank you for the suggestion. In the manuscript for the 1st round review, we summarized the GHG emissions of the study scenarios in Table 2. In this revised manuscript, we inserted a table displaying the peak annual and cumulative GHG emissions in Figures 2 and 4. Table 2 in the old manuscript was deleted.

Lines 231-232, the authors pose that higher average FC ICVs “commands” higher NEV sales volume—there is an inherent assumption throughout the results that indicates that ICV FC will be tied to NEV sales. It is important to note that this is only true if the CAFC regulation is a binding constraint—if automakers decide to violate the rule and pay a fine (unlikely) or if they exceed the credit requirements (more likely in scenarios with high NEV), there is no “guarantee” that they will automatically make the vehicles in the rest of their fleet “dirtier”. At the very least, this assumption should be noted in the text.

Thank you for your comments. It is true that automakers can decide to violate the rule. However, the Dual Credit policy sets some very strict penalties, so we do not see this as a major risk in our model. For example, noncompliant automakers must halt the production of a specified number of gas-guzzling models and submit and execute an adjusted production or importation plan. We assume that automakers will be unlikely to choose non-compliance if they are forced

to halt production of ICEV models that have fuel consumption that exceed target values. Thus, the most important assumption of the study is that automakers will meet the Dual Credit policy.

Since the objective function of the NEOCC model is to maximize the total gross profit of the industry, the model will converge to sell just enough NEVs and/or “low FC ICEVs” to meet the policy requirements. There is no mechanism to promote the sales of higher emissions ICEVs in the current NEOCC model. Exceeding the Dual Credit policy requirements is observed when NEVs are competitive enough. For example, under the “DC-Reference” scenario, this occurs starting 2039. Under the “DC-optimistic” scenario, the Dual Credit policy requirements are exceeded for all MYs after 2022. For these MYs, the market shares are determined based on the vehicle total cost of ownership (TCO) and the corresponding probabilities calculated through the discrete choice model (a nested multinomial logit model described by Figure S5).

We added the following sentences in the section “Design of scenarios” to better clarify the assumptions given in this study:

“In almost all scenarios, the Dual Credit policy is either achieved or exceeded for each model year (MY) to avoid penalties such as halting the production of high FC ICEVs. Two exceptions to this assumption include the PT-CAFC and EX-ICEV scenarios. In these scenarios, the Dual Credit policy is ignored, and policy targets are met with CAFC rules only. For simplicity, we do not consider credit carryover if surplus credits are generated in the previous MY.”

Line 285, “transportation policies should consider emissions based on the whole life cycle”—why? If NEVs/ZEVs’ upstream emissions must be accounted for by automakers, with what mechanism do these companies have to improve the grid emissions? Perhaps it would be better to state that transportation policies should be coupled with policies that clean the grid, giving the correct stakeholders some regulatory incentive to improve grid emissions (rather than OEMs which have no power in this domain).

Thank you for the comments and perspective on this sensitive issue. The reason we recommend considering the emissions of the whole vehicle life cycle is to avoid or minimize shifting the GHG emissions from the transportation sector to the power generation sector. This can be achieved by correctly counting the GHG emissions of electricity consumed by NEVs. The U.S. CAFE regulations calculate the petroleum equivalent fuel economy of BEVs using the petroleum equivalency factor (PEF), which is equal to 33,705 Wh/gal for BEVs without petroleum-powered accessories. Since the PEF considers the upstream efficiency of fuel production, the national average electricity generation, and transmission efficiencies, it is a regulation based on life-cycle emissions. In China, there is much discussion about implementing a similar methodology. Although not adopted in the Stage 5 CAFC regulations yet, it is very likely to be adopted in the future and worthy to be discussed. In this paper, the scenario “DC-BEVFC” is designed to explore the potential for reducing GHG emissions by adopting a life-cycle-based transportation policy. Thus, it is meaningful to discuss and compare the GHG emissions under the scenarios with PEF and without PEF. This study is able to quantify the differences of GHG emissions under these scenarios and help policymakers to better understand the potential impacts of the policies.

In addition, we also agree that the transportation policies should be coupled with other policies to reduce the GHG emissions of electricity generation. As seen in “S8. Uncertainties of GHG emissions,” the GHG emissions of light-duty passenger vehicles (LDPVs) are sensitive to the

GHG intensity of the electricity. This is not surprising as BEVs account for a significant market share after 2030 and their emissions are entirely dependent on the grid's carbon intensity. We believe it is possible to reduce GHG emissions by allowing carbon trading among different industries.

We added the following sentences in the revised paper to address the impact of the grid and policies affecting it:

“With an increasing NEV market share in the future, a cleaner electric grid will play a more significant role in reducing the life-cycle GHG emissions of the transportation sector. Thus, in the long term, transportation policies should be coupled with other policies to accelerate the decarbonization of the electric grid by offering stakeholders some regulatory incentives to reduce grid emissions.”

Line 294, “...calibrated with 2017 Chinese passenger vehicle market data...”, the vehicle choice model being calibrated to a single year appears to be relatively problematic. How do the authors account for changes in consumer preferences over time?

This is an excellent question/suggestion. We calibrated the new version of the NEOCC model with 2016–2019 Chinese passenger vehicle market data. The market information in 2016-2019 used for calibration is summarized and shown in Supplementary Table S4.

The consumer preference is quantified based on the vehicle total cost of ownership (TCO) and the probability through the discrete choice model (a nested multinomial logit model) as shown in Supplementary Figure S5. Over time, the TCOs for different vehicle technologies vary due to the changes in charging/refueling infrastructure, vehicle cost, energy/fuel cost, subsidies, and travel patterns (such as vehicle kilometers traveled). In this study, the travel patterns of consumers are assumed unchanged, while other influencing factors, such as the charging/refueling infrastructure, vehicle technologies, and subsidies, are assumed to evolve over time. For example, lowering battery cost reduces the TCO of BEV owners when other conditions are unchanged and, therefore, increases the probabilities of consumers purchasing BEVs.

Equation 1, why are profits being minimized? Shouldn't profits be maximized (or I suspect the authors were trying to minimize the negative profit).

Equations 2-4, these equations are nonsensical. C(x) and N(x) and not adequately explained. The equations read: “CAFC credits and NEV credits should be greater than or equal to 0”.

Thank you for pointing out the error in Equation 1. Yes, the profits should be maximized. We corrected Equation 1 in the revised paper. We also provided more explanation of C(x) and N(x) in the revised paper:

“The constraints of the Dual Credit policy include the NEV credit rules, as shown in Eqn. (2), the CAFC credit rules, as shown in Eqn. (3), and the combination of NEV credit rules and CAFC credit rules, as shown in Eqn. (4).

$$\text{Max}_x R(x) = P(x) - I(x) - M(x) \quad (1)$$

$$s. t. N(\mathbf{x}) \geq 0 \quad (2)$$

$$s. t. C(\mathbf{x}) \geq 0 \quad (3)$$

$$s. t. N(\mathbf{x}) + C(\mathbf{x}) \geq 0 \quad (4)$$

where, $R(x)$ in Eqn. (1) is the industry's total gross profit to be optimized and x is a vector that groups a set of decision variables (industry's internal subsidies) to realize this optimization. $P(x)$ is the sum of the products of vehicle prices by vehicle type and the corresponding sales volumes by vehicle type. $I(x)$ is the sum of the products of industrial internal subsidies by vehicle type and the corresponding sales volumes. $M(x)$ is the sum of the products of production cost by vehicle type and the corresponding sales volumes. $C(x)$ is the sum of the products of CAFC credits achieved by vehicle type and the corresponding sales volumes. $N(x)$ is the sum of the products of NEV credits achieved by vehicle type and the corresponding sales volumes. The calculations of CAFC credits and NEV credits for different vehicle types are explained by the Dual Credit policy 5. The sales volumes for these vehicle types are calculated through the discrete choice method in the NEOOC model. The algorithm logic flow is explained in Supplementary Figure S5. More detailed information is provided in studies by Ou et al ^{18,24,25.}

Lines 358-368, considerably more effort should be placed in the forecasts of the electricity grid. It is unclear how robust the results are to the changing electricity grid, but at a minimum these should be tested at the same level of robustness/detail as scenarios of CAFC and NEV given how likely the grid scenarios will be critical to the final results of the paper.

Thank you for the suggestions, we agree we should be clearer on the forecasts of the grid. The WTW GHG emissions of BEVs are entirely dependent on the GHG intensity of the electricity grid. In this paper, we added a dedicated section in the main text to discuss the impact of the GHG intensity of the electric grid. We considered three electricity grid scenarios: (1) the Reference Technology Scenario (RTS) defined in IEA's Energy Technology Perspectives were used as the reference scenario; (2) the EIA 6 °C scenario was used to represent business-as-usual conditions; and (3) the 2 °C scenario was used to represent the highly challenging scenario. Figure 4 demonstrates the uncertainties due to the electricity GHG intensity for four selected scenarios. For other scenarios, the GHG uncertainties due to the GHG intensity of the electric grid are presented in Supplementary Material S8.

Impact of the GHG intensity of the electric grid

With increasing market shares of BEVs and PHEVs, the GHG emissions related to electricity consumption play a more significant role. Thus, it is important to assess the uncertainties and robustness of the results due to the changing electric grid. The Reference Technology Scenario (RTS) defined in IEA's Energy Technology Perspectives ²³ is adopted as the reference (RefEle) for the changes in electricity mixes of China. IEA's RTS factors in current commitments are endorsed by the Chinese government (as well as other countries) to limit GHG emissions and improve energy efficiency. These commitments are projected to help limit the global long-term temperature rise to 4 °C. IEA also designed 6 °C and 2 °C scenarios to represent business-as-usual and highly challenging scenarios, respectively, for the global energy sector that would achieve the long-term average global temperature rises of 6 °C and 2 °C. They are used in this study to calculate the upper (HighEle) and lower (LowEle) limits of GHG intensities of electricity. The

well-to-wheels (WTW) GHG intensity of the Chinese electric grid is estimated as 185 g/MJ in 2018. With the continuous penetration of the renewable (wind, solar, etc.) and nuclear power generation technologies, the GHG intensities of electricity in 2040 are projected to be 108, 142, and 52 g/MJ for the RefEle, HighEle, and LowEle scenarios, respectively.

Figure 4 demonstrates the uncertainties due to the electricity GHG intensity for four selected scenarios. The uncertainties of all other scenarios are provided in Supplementary Material S8. As expected, the uncertainties increase over time due to the higher NEV market share and the increasing differences in electricity GHG intensities between the three electric grid scenarios. A cleaner grid not only decreases overall GHG emissions but also advances the peak GHG date. For example, under the DC-Ref-LowEle scenario, the peak annual GHG emissions occur in 2029, which is 3 years before the DC-Ref-RefEle scenario. The Ex-50%BEV scenario has the highest uncertainty because it has the highest BEV market share, while the EX-ICEV scenario exhibits the lowest because of low exposure to the grid. Under the LowEle scenario, the annual GHG emissions of the DC-BEVFC scenario could be lower than the EX-ICEV scenario after 2038. This clearly shows the importance of policy design and a clean electric grid in reducing GHG emissions.

Figure 1. Annual life-cycle GHG emissions of the LDPV fleet in China under the DC-Reference, EX-50%BEV, EX-ICEV, and DC-BEVFC scenarios. The shaded area represents the uncertainties due to the GHG intensity of the Chinese electric grid. *The percentage in parenthesis represents the difference between that scenario and the DC-Ref-RefEle scenario.

Reviewer #3:

This paper addresses a timely policy issue that is-- the linkage and interactions between China's CAFC and NEV Mandate could derail the efforts to improve ICEVs efficiency and therefore increase GHG emissions. By using different scenarios, especially the two extreme ones, the authors highlighted the severity of the drawbacks in the designing of China's Dual-Credit Policy.

The analysis is solid and scenarios are carefully designed. The two models undergird this paper are well-developed and well-known.

I have three issues with this paper, but they in no way affect the overall efficacy of this paper.

Thank you for reviewing this paper and providing constructive suggestions. We also appreciate the supportive comments.

Since the first submission of the paper to Nature Communications in February, we have been working on model refinement with the China Automotive Technology and Research Center (CATARC), a leading Chinese organization that developed the Dual Credit policy. We also took your suggestions into considerations on the model revisions. Below are the major updates we made to the NEOCC model for this paper:

- (1) Recalibrated the NEOCC model with 2016-2019 Chinese passenger vehicle market data.
- (2) Considered full-hybrid ICEVs only as the "ICEV Low FC" technology which qualifies for the "Low Fuel Consumption" credit in the NEOCC model. The major consequence is the reduction of market shares in "ICEV Low FC". The micro/mild-hybrid ICEVs are included into the "ICEV Avg FC" technology, which helps reduce the FC of "ICEV Avg" at a faster rate. The classifications of "ICEV Low FC" and "ICEV Avg FC" are given by Table S4 in the Supplementary Material S5.2.
- (3) Updated the vehicle production cost based on the 2016-2019 vehicle sales and MSRP data provided by CATARC. The paper discussed the detailed cost analysis of the Chinese vehicle market that was just published by Energies [1].
- (4) Reduced the reference battery production cost from \$116/kWh to \$104/kWh by 2030 to reflect the recent battery cost reduction trend in China. Overall, BEVs become more competitive in China.
- (5) Considered the efficiency improvement of BEVs, which reduces the battery size required for a specific driving range.
- (6) Reduced the projection of gasoline price in 2020 to reflect the COVID-19 impact.

Some simulation results shown in this revised manuscript are slightly changed as the model and assumption were revised. However, these variations do not change the general conclusions of the paper. Following summarizes the two major changes:

1. The market share of "ICEV Low FC" technology in the updated model version is smaller than it was calculated in the older model version, because the "ICEV Low FC" technology is calibrated to full-hybrid ICEVs only in the new model version.
2. In the new model version, the NEV market share shows a gradual increase starting 2020 under the "PT-CAFC" scenario. However, in the older model version, the market share of NEVs maintains at about 8% before 2026, and then quickly increases to 24% by 2030. This change is mainly due to the item (2) mentioned above. Setting a more stringent

“ICEV Low FC” requirement increases the production volumes of “ICEV Low FC”. Meanwhile, the lower battery cost assumption (item (4)) also improves the competitiveness of NEVs. As a consequence, we observe a smoother increase in the NEV market share.

1) The assumption that Chinese cars' real CAFC compliance. As demonstrated in the VW Diesel-gate and many of the subsequent studies, new car CAFC compliance could be problematic, China included. On top of that, as auto components deteriorate, the gap between the CAFC target and the real FC gets bigger. One of the arguments for promoting BEVs is that relatively speaking BEVs tend to have long-life reliability in terms of energy consumption. So there could be a 10-20% gap between the real world CAFC and the "paper" CAFC readings in car's life time. I hope the authors could address this issue.

Thank you for the comments. We completely agree with you that the gap between the real-world CAFC and theoretical CAFC needs to be considered. This is especially important for BEVs, which tend to suffer higher electricity consumption rates (ECRs) under cold temperatures and due to battery degradation. In the present study, we recognize the gap between the labeled and real-world fuel consumption rates (FCRs) of vehicles. Therefore, we included a labeled-to-real-world adjustment in the China Vehicle Fleet Model to reflect the corresponding impacts. Based on previous research results, we adjusted the real-world FCRs/ECRs to the labeled values by multiplying by 1.2 for ICEVs [1,2] and 1.4 for BEVs [3-5].

Reference:

1. Huo, H., Yao, Z. L., He, K. B., Yu, X. Fuel consumption rates of passenger cars in China: Labels versus real-world. *Energy Policy*, 39 (11), 7130-7135, 2011.
2. Lu, Z.; Zhou, Y.; Cai, H.; Wang, M.; He, X.; Przesmitzki, S. *China Vehicle Fleet Model: Estimation of Vehicle Stocks, Usage, Emissions, and Energy Use-Model Description, Technical Documentation, and User Guide*; Argonne National Laboratory: Lemont, IL, 2018.
3. Zhou, Y., Vyas, A. *VISION model description and user's guide: model used to estimate the impacts of highway vehicle technologies and fuels on energy use and carbon emissions to 2100*. Argonne National Laboratory: Argonne, IL, USA, 2014.
4. Elgowainy, A., Han, J., Poch, L., Wang, M., Mahalik, M., Rousseau, A. *Well-to-wheels analysis of energy use and greenhouse gas emissions of plug-in hybrid electric*. ANL/ESD/10-1, June, 2010.
5. Stephens, T., Zhou, Y., Elgowainy, A., Duoba, M., Vyas, A., Rousseau, A. *Estimating on-road fuel economy of PHEVs from test and aggregated data*. Proceedings of 92nd Transportation Research Board Annual Meeting. Washington, DC, USA. 13-4755, 2013.

The following sentences are added in the “Methods” section of the revised paper:

“In the present study, we recognize that there are gaps between the vehicle labeled and real-world fuel consumption rates (FCRs) and include a labeled to real-world adjustment in the China-Fleet model to reflect the corresponding impacts. Based on previous research results, we adjust the labeled FCRs/ECRs to the real-world ones by multiplying a ratio of 1.2 for ICEVs^{26,30} and 1.4 for BEVs^{29,31,32}”

2) In assuming NEV credit requirement. The authors correctly mentioned that the credit requirement in 2023 is 18%, however, the authors then assumed that in 2030, the NEV credit requirement would be 40% (supposedly to match the 40% NEV market-share target). This is wrong. Assuming the average range of all the NEVs in 2023 is 200 km, then meeting 18% of NEV requirement would only need 11.3% of NEVs in total market share in 2023. If we assume the same conversion rate, then to reach 40-50% market share in 2030, China's NEV credit requirement would need to raise to 64-80% of NEVs with the same 200 km BEVs (and PHEVs). In short, NEV Credit requirement is larger in percentage than in vehicle numbers.

Thank you for the comment. You are right, the credit requirement is not based on market share. We also need to consider the number of credits granted to each NEV. In the NEOCC model, we assume that the future Dual Credit rules will be designed to promote BEV efficiency instead of long electric driving range. As such, we assume the number of NEV credits granted to each NEV decreases linearly to 1.0 by 2030.

The assumption of a 40% NEV quota is not to match the 40% NEV market-share target. In order to make reasonable assumptions for future NEV quotas, we conducted a sensitivity analysis by creating three scenarios with the NEV quota set at 30%, 40%, and 50% by 2030. Comparing the simulation results of these three scenarios, we found no significant differences in the BEV market share or total GHG emissions. We also found that achieving a CAFC target of 3.7 L/100 km (Worldwide harmonized Light vehicles Test Cycles (WLTC)-based FC, adjusted by the sales and vehicle curb weight) by 2030 is a much more challenging task than meeting the NEV quota (whether it is 30%, 40% or 50%). The NEOCC model shows that the CAFC rules are the driving force that influences NEV market share before 2030. Although a higher NEV quota (50%) reduces the surplus NEV credits available to compensate for the deficits of CAFC credits, a higher NEV quota does not significantly increase the market shares of vehicles categorized as NEV and ICEV Low FC. This finding is consistent with our previous results published in 2018, which have shown that it is easier for the automotive industry to meet the NEV quota than to meet the CAFC targets and that NEV credits are often used to make up the CAFC deficits (<https://doi.org/10.1016/j.enpol.2018.06.017>).

A higher NEV quota increases the cost of industry's internal subsidies to further promote the sales of NEV and Low-FC ICEVs. Although the higher subsidies can increase NEV market share to meet the requirements, the higher subsidy cost reduces industry profits. Under the reference Dual Credit scenario with a 40% NEV quota, the industry profit is 30% lower than the 2019 level.

In addition, after evaluating a series of government policies published in recent years and discussing the issue with experts and policymakers in China, we believe 40% is a reasonable target for the 2030 NEV quota used in the reference Dual Credit scenario.

Furthermore, since no NEV market share targets for years after 2030 have yet been discussed in the official documents, we also assume this 40% NEV quota will remain in place beyond 2030.

3) Line 168, "the BEV and PHEV market shares increase steadily to 21.5% and 14.2% by 2030." PHEVs share assumed here is too big. In 2019, the share of PHEVs in total PEVs was only 21%. Globally, PHEVs are out of fashion. GM stopped selling Volt. A reasonable assumption would be 5-10% of the total PEVs in 2030.

Thank you for the comment. We agree with your assertion that PHEVs are becoming out of fashion for mainstream applications. PHEV employ two powertrain systems, which is costly and technically challenging.

The PHEV market share has been updated using a newer version of the NEOCC model, which implemented the following modifications:

- (1) Calibrated the NEOCC model with market data for 2016–2019 (the previous model version was calibrated with 2017 market data only). We found that the BEV battery cost drops at a faster rate, making BEVs increasingly more competitive.

- (2) Reduced the reference battery production cost from \$116/kWh to \$104/kWh by 2030 to better reflect the recent battery cost reduction trend in China.
- (3) Considered the improvement in the electricity consumption efficiency of BEVs, which can potentially reduce the battery size required for a given driving range. Both adjustments improved the competitiveness of BEVs in the model.

Under the reference Dual Credit scenario, the new simulation results show that the BEV and PHEV market shares will steadily increase to 22.6% and 9.3%, respectively, by 2030. In the near future, the PHEV with a small engine and a range extender may find its place in mountain areas and for people who travel long distances, so it still has a role in the vehicle market. The Chinese domestic automakers, such as SAIC, Geely, and Chang'an, still consider PHEVs strategically important to meet Dual Credit policy requirements.

In addition, under the reference Dual Credit scenario, the new simulation results also show that the market share of PHEVs will shrink to 6.2% by 2040, after they reach their peak (9.3%) in 2030. Compared to BEVs, we also believe the market share of PHEVs will be limited in the long-term future.

In conclusion, this is a very good paper. I would love to hear the authors' responses.

Again, we really appreciate the constructive suggestions and supportive comments.

REVIEWERS' COMMENTS:

Reviewer #2 (Remarks to the Author):

The responses to the previous comments have been well documented and addressed. With the updates, I believe the quality of the paper has significantly improved, and the majority of issues I found with the original manuscript have been accounted for.

I have two minor things to note:

-Fix line 138: broken reference

-In the response to reviewers, it was noted that cumulative emissions were added to Figures 2 and 4. There are no cumulative emissions in Figure 2 but I believe the authors were referring to Figures 3 and 4 (not 2), if this is not the case then the figure should be amended to the authors' intent.

Reviewer #3 (Remarks to the Author):

I appreciate authors' detailed revision and I am happy that my questions have been generally answered. The supplementary materials help me understand the underlining assumptions of the modelling and build a strong case for the separation of China's NEV credit policy and its CAFC policy.

I congratulate authors for their successful efforts to explain and demonstrate their research findings.

Response Letter to Reviewer Comments

The authors would like to thank the efforts of the editorial personnel and the reviewers, especially during this pandemic period. Comments are in black. Our responses are in blue.

Reviewer #2 (Remarks to the Author):

The responses to the previous comments have been well documented and addressed. With the updates, I believe the quality of the paper has significantly improved, and the majority of issues I found with the original manuscript have been accounted for.

Thank you for reviewing this paper and providing constructive suggestions. We also appreciate the supportive comments.

I have two minor things to note:

-Fix line 138: broken reference

Thank you for pointing it out. We corrected the broken reference.

-In the response to reviewers, it was noted that cumulative emissions were added to Figures 2 and 4. There are no cumulative emissions in Figure 2 but I believe the authors were referring to Figures 3 and 4 (not 2), if this is not the case then the figure should be amended to the authors' intent.

Thank you for finding the typo in the manuscript. You are right, it should be Figures 3 and 4. The manuscript has been updated.

Reviewer #3 (Remarks to the Author):

I appreciate authors' detailed revision and I am happy that my questions have been generally answered. The supplementary materials help me understand the underlining assumptions of the modelling and build a strong case for the separation of China's NEV credit policy and its CAFC policy. I congratulate authors for their successful efforts to explain and demonstrate their research findings.

Thank you for reviewing this paper and providing constructive suggestions. We also appreciate the supportive comments.